# Carcinoid Syndrome: Preclinical Models and Future Therapeutic Strategies

**DOI:** 10.3390/ijms24043610

**Published:** 2023-02-10

**Authors:** Giovanni Vitale, Silvia Carra, Ylenia Alessi, Federica Campolo, Carla Pandozzi, Isabella Zanata, Annamaria Colao, Antongiulio Faggiano

**Affiliations:** 1Department of Medical Biotechnology and Translational Medicine, University of Milan, 20122 Milan, Italy; 2Laboratory of Geriatric and Oncologic Neuroendocrinology Research, IRCCS, Istituto Auxologico Italiano, 20100 Milan, Italy; 3Laboratory of Endocrine and Metabolic Research, IRCCS, Istituto Auxologico Italiano, 20100 Milan, Italy; 4Endocrine Unit, University Hospital “Gaetano Martino” of Messina, 98125 Messina, Italy; 5Department of Experimental Medicine, Sapienza University of Rome, 00161 Rome, Italy; 6Section of Endocrinology and Internal Medicine, Department of Medical Sciences, University of Ferrara, 44121 Ferrara, Italy; 7Department of Clinical Medicine and Surgery, University of Naples Federico II, 80138 Naples, Italy; 8Endocrinology Unit, Department of Clinical and Molecular Medicine, Sant’Andrea Hospital, ENETS Center of Excellence, Sapienza University of Rome, 00189 Rome, Italy

**Keywords:** carcinoid syndrome, preclinical models, neuroendocrine tumors, serotonin, xenograft, pharmacological treatment

## Abstract

Carcinoid syndrome represents a debilitating paraneoplastic disease, caused by the secretion of several substances, occurring in about 10–40% of patients with well-differentiated neuroendocrine tumors (NETs). The main signs and symptoms associated with carcinoid syndrome are flushing, diarrhea, hypotension, tachycardia, bronchoconstriction, venous telangiectasia, dyspnea and fibrotic complications (mesenteric and retroperitoneal fibrosis, and carcinoid heart disease). Although there are several drugs available for the treatment of carcinoid syndrome, the lack of therapeutic response, poor tolerance or resistance to drugs are often reported. Preclinical models are indispensable tools for investigating the pathogenesis, mechanisms for tumor progression and new therapeutic approaches for cancer. This paper provides a state-of-the-art overview of in vitro and in vivo models in NETs with carcinoid syndrome, highlighting the future developments and therapeutic approaches in this field.

## 1. Introduction

Small intestinal neuroendocrine tumors (NETs) occur in about 1 out of every 100,000 people [1]. Approximately 10–40% of patients with well-differentiated NETs develop carcinoid syndrome (CS), a debilitating paraneoplastic disease caused by the secretion of several humoral factors, significantly worsening the patient’s quality of life and prognosis [2,3]. This syndrome is characterized by flushing, diarrhea, abdominal pains, hypotension, tachycardia, bronchoconstriction, venous telangiectasia, dyspnea and fibrotic complications, such as mesenteric and retroperitoneal fibrosis, and carcinoid heart disease (CHD) [4,5]. CS is mostly associated with liver metastases in patients with midgut NETs, but it may also occur in bronchial, pancreatic and other NETs [4]. In the U.S. the number of NEN patients with CS significantly increased from 50 (11%) of 465 patients in 2000 to 160 (19%) of 854 in 2011 [6]. Between 20–70% of patients with CS develop CHD, but the overall prevalence is uncertain due to a lack of consistent screening using transthoracic echocardiography [2,7,8,9]. The occurrence of CHD is associated with a poor prognosis. Indeed, mortality was significantly higher in patients with CHD (40%) compared with NETs without CHD (25%) [10].

Several substances have been identified as being potentially related to CS manifestations and include mostly amines (serotonin, 5-hydroxytryptophan, norepinephrine, dopamine, histamine), but also many different polypeptides (kallikrein, pancreatic polypeptide, bradykinin, motilin, somatostatin, insulin, S-100 protein, vasoactive intestinal peptide, neuropeptide K, substance P, neurokinins, ACTH, gastrin, growth hormone, peptide YY, glucagon, endorphins, neurotensin, chromogranin A) and prostaglandins [11]. The relative impact of any of them for triggering particular symptoms and complications of CS remains unclear, but serotonin (5-HT) and metabolites from the 5-HT pathway seem to play a crucial role [12].

Actually, there are two main approaches in the management of CS: the direct inhibition of hormonal secretion and the control of tumor burden, thereby reducing the disease volume and indirectly the hormonal production. The first approach is represented by the use of somatostatin analogues (octreotide, lanreotide and pasireotide), which provide rapid symptom relief [13]. Telotristat ethyl, a tryptophan hydroxylase (TPH) inhibitor, the rate-limiting enzyme in 5-HT biosynthesis, represents a novel treatment in patients with CS [14]. On the other hand, approaches aimed at controlling tumor growth, such as chemotherapy, type I interferons, peptide receptor radionuclide therapy, tyrosine kinase inhibitors or liver-directed therapies (radiofrequency ablation, cryoablation, transarterial embolization, chemoembolization and radioembolization), give a more delayed mode of action in symptoms control (with the exception of liver-directed therapies) [15,16].

Although there are several drugs available for the treatment of CS, the lack of therapeutic response, poor tolerance or resistance to drugs are often reported. Preclinical models are indispensable tools for investigating the pathogenesis, mechanisms for tumor progression and new therapeutic approaches for cancer. This paper provides a state-of-the-art overview of in vitro and in vivo models in NETs with CS and highlights the future developments and therapeutic approaches in this field.

## 2. Preclinical Models of Carcinoid Syndrome

### 2.1. In Vitro Models

In the last decades, there is growing interest in the development of in vitro models for NETs. Primary cultures of human NETs are difficult to be established and maintained, due to several factors such as the small amount of tissue available, low mitotic activity and low adhesion to plastic [17]. Tumor 3D cultures have been recently adopted as an alternative strategy to culture tumor cells from surgically removed NETs, as spheroids in extracellular matrix able to mimic the tumor micro-environment [18].

In order to overcome limitations related to the isolation of primary cultures, several immortalized NET cell lines have been established. The most used and well-characterized NET cell lines are listed in Table 1.

The first human carcinoid cell line was established by Debons-Guillemin in 1982. These cells (named CGP cells) were obtained from a 29-year-old male with jejunal carcinoid tumor that presented high serum levels of 5-HT and histamine and high urinary levels of 5-HT and 5-hydroxyindole-acetic acid (5-HIAA). CGP cells display a very low proliferation rate and are able to synthesize, store and release both 5-HT and histamine [19].

Two years later, Kirkland and Ellison established a NET cell line from a 46-year-old male with a diagnosis of primary bronchial carcinoid tumor and high urinary 5-HIAA levels [50]. These cells display corticotropin-releasing factor-like activity after more than 100 passages [50].

The next successful cell line established from a 75-year-old male patient with a diagnosis of malignant carcinoid of the small intestine was the KRJ-I cell line [21]. These cells display classical morphological and immunocytochemical features of a small intestine carcinoid with similar characteristics to the naive human enterochromaffin cells, except for a marked resistance to octreotide-mediated 5-HT secretion inhibition [20]. They also secrete high levels of 5-HT, noradrenaline and pituitary adenylate cyclase and express chromogranin A (CgA), neuro-specific enolase, Ki-67, TPH-1, substance P and guanylin [22,23,24].

Evers and colleagues established and characterized BON-1 cell line, isolated from metastatic lymph node of a 28-year-old man with pancreatic carcinoid tumor [27]. These cells produce 5-HT, CgA, neurotensin, pancreastatin and display receptors for gastrin, somatostatin, 5-HT and acetylcholine [27]. A comprehensive cytogenetic profile of BON-1 demonstrated that this cell line harbors both numerical and structural genomic alterations indicative of endocrine pancreatic tumors [29]. BON-1 cells represent the best characterized and used in vitro experimental model for drug testing in NETs. A dose dependent cytotoxic effect of the tyrosine kinase inhibitor imatinib on these cells was reported [34]. Moreover, the immunosuppressor leflunomide and its metabolite teriflunomide have been demonstrated to be able to inhibit BON-1 cells proliferation and to induce G2/M phase arrest [26]. Lithium treatment caused a dose-dependent reduction in BON-1 growth [28] as well as sodium butyrate and hexamethylene bisacetamide [30] and all-trans-retinoic acid [25]. These cells are known to exhibit constitutively activated phosphatidylinositol-3-kinase (PI3K)/Akt/mammalian target of rapamycin (mTOR) signaling. The rapamycin analogue everolimus potently inhibited their growth, induced G0/G1-phase cell cycle arrest as well as apoptosis [35]. Moreover, treatment with the PI3K inhibitor LY294002 induced significant cell growth inhibition and a decrease in CgA levels [31], while treatment with the Akt inhibitor MK-2206 significantly reduced cell proliferation in a dose-dependent manner [32]. A comparative study demonstrated that the multiligand somatostatin analog (SSA) pasireotide inhibited 5-HT secretion more potently than octreotide, which may be explained by the relative high expression of Somatostatin Receptor Type (SST) type-5 (SST-5) in this cell line [33]. The IGF pathway plays an important role in autocrine/paracrine growth of BON-1 cells and interferon β has been demonstrated to highly inhibit their proliferation via IGF suppression [38,39].

A more recent human pancreatic neuroendocrine cell line is QGP-1, derived from a 61-year-old man with a diagnosis of pancreatic islet cell carcinoma. These cells highly express enterochromaffin cell marker genes, such as TPH-1, CgA, synaptophysin, ATP-dependent vesicular monoamine transporter 1 (VMAT1), metabotropic glutamate receptor 4, b1-adrenergic receptor, muscarinic 4 acetylcholine receptor, substance P, 5-HT transporter, guanylin and somatostatin [33,41]. QGP-1 cells are sensitive to everolimus and SSA as well as to the PI3Kα inhibitor BYL719 [36,40,51].

SA Foregut NETs are both phenotypically and genotypically distinct from midgut carcinoid tumors. CNDT2 has been reported as a midgut carcinoid cell line [42]. However, several investigators have questioned the authenticity of its neuroendocrine background, and the origin from a patient with ileal carcinoid has not been confirmed. Therefore, this cell line should not be used as a model for CS [52].

NCI-H727 is a typical bronchial carcinoid cell line derived from a 65-year-old female patient. These cells express detectable levels of p53 mRNA and are able to secrete a bioactive parathyroid hormone-like protein, while their growth is inhibited by EGFR monoclonal antibodies and BYL719 [36,43]. The treatment of NCI-H727 cells with the PI3K inhibitor LY294002 significantly reduced tumor cell growth [44]. NCI-H727 cells expressed SST2- and 5 subtypes and SSA inhibited their growth [42].

The same SST subtypes were expressed by NCI-H720 cells, the best characterized human atypical bronchial carcinoid cell line, whose growth can be inhibited by several cytotoxic agents, i.e., the quinone-based small molecule compound NSC 95397, the macrocyclic lactone brefeldin A and the proteasome inhibitor bortezomib [46]. A recent study demonstrated that NCI-H720 cells are sensitive to the SST-5 analog BIM23206 displaying a similar sensitivity to lanreotide and the SST-2 analog BIM23120, while NCI-H727 are responsive to BIM23120 and to the pan-SST analog BIM23A779 [23,45].

Other CS cell lines include COLO 320DM cells, derived from a human colon carcinoid secreting 5-HT, parathyroid hormone (PTH), adrenocorticotropic hormone (ACTH), norepinephrine and epinephrine [47], and GOT1 cells, established from a liver metastasis of a human ileal carcinoid tumor, expressing all recognized somatostatin receptors, the amine transporters VMAT1 and ATP-dependent vesicular monoamine transporter 2 (VMAT2) [49].

Even if several cell lines representative of CS are currently available, further studies evaluating the neuroendocrine characteristics of these cells and their therapeutic sensitivity should be performed to improve their choice and management.

### 2.2. In Vivo Models

In literature there are not many preclinical in vivo studies on CS (Table 2). In vivo models adopted for CS can be summarized in three main groups: (1) xenotransplantation of NET cell lines in nude mice, (2) administration of 5-HT in murine and rabbit models or transgenic murine models with alterations of 5-HT pathway, and (3) mice models of intestinal inflammation.

Kölby L. et al. and Bernhardt P. et al. conducted preclinical studies using GOT1 cell line-bearing mice. The GOT1 cell line was successfully transplanted into nude mice and propagated for five consecutive generations. Tumor cells were similar to the original tumor, expressing SSTs and VMAT1 and VMAT2, and secreting 5-HT and its metabolite 5-HIAA [49,67]. Kolby L. et al. demonstrated that in GOT1-bearing nude mice, all xenografted tumors could be visualized by scintigraphy using the SSA 111Indium (111In)-octreotide and Iodine-123 metaiodobenzylguanidine (123I-MIBG), due to the expression of SSTs and VMAT1 and VMAT2. For this reason, GOT1-bearing nude mice represent a potential model for the study of midgut carcinoid tumors [49]. Bernhardt P. et al., furthermore, proposed a therapeutic protocol for tumor reduction in GOT1-bearing nude mice [53]. They administered first radiolabeled injections of 177Lutetium-[DOTA°,Tyr3]octreotate (177Lu-DOTATATE) and then 111In-DOTATATE, demonstrating that suboptimal therapeutic amounts caused an increased uptake of the second injection [53].

Musunuru S. et al. injected into 17 nude mice the human pancreatic carcinoid BON-1 cell line and showed that 65% of them developed carcinoid liver metastases with elevated 5-HT levels and a typical fibrosis of the valvular tissue, above all on tricuspid valve, which resembles CHD observed in patients [54].

Jackson L.N. et al. proposed an in vivo model of CS obtained through the injection of BON-1 cells into male athymic nude mice’s spleen [55]. Most of these animals developed liver metastasis with positive staining for 5-HT and CgA, increased 5-HT plasma levels and 5-HIAA urine levels, and the development of mesenteric fibrosis, diarrhea and fibrotic cardiac valvular disease (tricuspid and mitral thickening), reminiscent of CS. In addition, mice injected with BON-1 cells and treated with octreotide, a SSA, or bevacizumab, a Vascular Endothelial Growth Factor (VEGF) inhibitor, developed fewer liver metastases and manifestations of CS, including valvular heart disease, compared to untreated control [55]. More specifically, valvular heart disease appears to be a consequence of chronic effects of 5-HT release on the heart with the development of carcinoid plaques that contain subendocardial deposits of myofibroblasts, fibroblasts and smooth muscle cells in a myxoid matrix [68].

A similar model has been adopted implanting NCI-H727 cells in nude mice to evaluate the antitumor activity of peptide receptor radionuclide therapy and everolimus. However, no information is available on the development of fibrosis and diarrhea and increased production of 5-HT in animals after the implantation of NCI-H727 cells. Therefore, its use as a model for CS has been not confirmed yet [69,70,71].

The origin of CHD remains unknown but among patients with CS, those with heart disease have higher levels of tachykinin and 5-HT in serum and 5-HIAA in urine [72]. Interestingly, Sari Y. et al. demonstrated that 5-HT and 5-HT receptors (5-HTR) play a crucial role in normal heart embryogenesis and that cultured heart cells express 5-HT transporter (5-HTT) [73]. On the basis of this evidence, several preclinical studies reproduced the 5-HT hypersecretion, typical of patients with CS, in murine models. Gustafsson B.I. et al. conducted the first case control preclinical study on rats based on the long-term 5-HT administration. In rats, 5-HT injections induced clinical signs observed in patients with CS, such as flushing, loose stools and anorexia. After three months, cardiac disease with pathological echocardiographs and histopathological changes, similar to those observed in the CHD, appeared. Histopathological examination of the hearts in the 5-HT group revealed shortened and thickened aortic cusps with an increased cellularity of myofibroblasts in a collagenous matrix [56].

Elangbam C.S. et al. demonstrated that subcutaneous injections of 5-HT for seven days resulted in thickening and compositional alteration of aortic and mitral valves in Sprague–Dawley rats, due to the higher amount of glycosaminoglycans and a reduction in collagen, respectively, caused by the up- and down-regulation of 5-HT_2B_ receptor (5-HT_2B_R) and 5-HTT genes [57]. Lancellotti P. et al. confirmed in a rabbit model that long-term oral 5-HT overload can cause valvular heart disease, similar to that reported in patients with CHD [58].

Regarding the 5-HT_2B_R, Nebigil C.G. et al. generated transgenic mice over-expressing this receptor specifically in the heart, resulting in compensated hypertrophic cardiomyopathy associated with proliferation of the mitochondria. This evidence suggests a role for mitochondria in the hypertrophic signaling that is regulated by 5-HT [59]. On the other hand, 5-HTT is a key protein responsible for 5-HT clearance and cellular internalization of 5-HT; it is highly expressed in platelet and in the pulmonary artery endothelial and smooth muscle cells of the lung [60]. Interestingly, 5-HTT-deficient (5-HTT-KO) mice developed structural and/or functional cardiac abnormalities and valvulopathies simulating human CHD. The absence of this transmembrane protein may result in increased and persistent interactions between 5-HT and 5-HTR and valvular mitogenic activity with extracellular matrix production in the 5HTT-KO mice [60]. All these studies demonstrated that 5-HT most likely is involved in the pathogenesis of CHD. Most of these models have also been used to study the effects of several compounds able to inhibit the synthesis of 5-HT or to antagonize 5-HTR, as described in the section related to the future therapies.

Moreover, Contractor T. et al. proposed an innovative genetically engineered mouse model (GEMM) which generates ileal NETs: the transgenic RT2 mice is considered a classic model for pancreatic NETs, but its B6AF1 genetic background is associated with the development of ileal NETs and the secretion of 5-HT in at least 22% of these tumors. Authors discovered that loss of imprinting of IGF2 occurred in 57% of patients with ileal NET and observed that RT2 mice in a B6 genetic background with genetically increasing IGF2 activity developed ileal NETs. For this reason, IGF2 may be considered the first genetic driver of ileal neuroendocrine tumorigenesis [64], followed by the loss of copy of MIR1-2 and attenuated RB activity [74].

Lastly, several mice models of intestinal inflammation, obtained through both dextran sulfate sodium (DSS)-induced colitis and infection with Trichuris muris, have been adopted to test the effects of several TPH inhibitors clinically used in the therapy of CS [65,66].

## 3. Future Therapies for Carcinoid Syndrome

The 5-HT biosynthesis is catalyzed by TPH of which there are two isoforms: TPH-1, expressed in enterochromaffin cells of the gastrointestinal tract, and TPH-2, that is considered to be the source of the central neurotransmitter pool of 5-HT and predominantly expressed in raphe neurons of the brainstem and myenteric neurons in the gut. The majority of blood 5-HT is synthesized by TPH-1 that emerged as a possible therapeutic target for the development of specific inhibitors [75]. The first drugs studied in vivo were p-chlorophenylalanine (fenclonine, PCPA) and p-ethynylphenylalanine (PEPA), which reduced blood 5-HT and 5-HIAA urine levels in Wistar rats [62]. Both drugs showed a restricted therapeutic potential because of central nervous system-mediated side effects related to the non-selective inhibition on isoform of TPH [75]. In mice, Margolis et al. reported that oral administration of TPH inhibitors (LP-920540 and telotristat) is able to reduce 5-HT levels in the intestinal mucosa [65]. Kim et al. not only confirmed the previous evidence in mice models of intestinal inflammation but also showed improvements in bowel movement frequency with consequent attenuation of colitis severity and diarrhea episodes, reducing stool frequency and consistency [66]. Therefore, preclinical data supported the potential utility of telotristat in CS, particularly in the treatment of diarrhea [76].

In March 2017, telotristat ethyl was the first-in-class, small-molecule, peripheral TPH-1 inhibitor, which does not cross the blood–brain barrier, to be approved by the FDA for the management of CS-related diarrhea refractory to somatostatin analogs. Since the exclusion from the brain may not be complete, it would be necessary to develop novel compounds not inhibiting TPH-2, in order to avoid adverse events [75]. Although telotristat ethyl has been approved in this clinical contest with many clinical benefits, several side effects were evident, indicating a continued need for safe efficacious drugs for this indication [76]. A recent phase III non-randomized study evaluated the long-term safety and tolerability of telotristat ethyl. Although no event occurred in more than eight patients (6.5% of the study population), the most common were gastrointestinal disorders (diarrhea, nausea, abdominal pain and constipation). Other related adverse events occurring in >2 patients included fatigue, decreased appetite, depression, asthenia, peripheral edema and increased hepatic enzyme. Fortunately, most adverse events were mild to moderate in severity [77].

Regarding CHD, no studies evaluated the effect of telotristat on this complication. Herrera-Martinez A.D. et al. showed that telotristat decreased 5-HT production in a dose-dependent manner without affecting cell proliferation of BON-1 cells [78]. However, they did not evaluate the anti-fibrotic activity of this drug. Future studies should be conducted in BON-1 xenograft model in order to assess the potential therapeutic effect of TPH inhibitors on CHD. Indeed, BON-1 xenograft model recapitulates in vivo human CHD, with elevated 5-HT levels and the development of valvular fibrosis. Interestingly, telotristat is currently the subject of a phase III clinical trial which aims to evaluate its role, combined with that of SSA, in controlling CHD in patients with metastatic NETs “https://clinicaltrials.gov/ct2/show/NCT04810091 (accessed on 10 January 2023)”.

A recent publication claimed to have discovered a selective inhibitor of TPH-1, mol002291, a molecule derived from the Chinese herb *Rheum officinale*. This agent specifically inhibited in vitro TPH-1, reduced colonic 5-HT content and attenuated visceral hyperalgesia in rat models [79]. Further studies should analyze the effects of mol002291 on the development of CHD in BON-1 xenograft models [76].

Concerning the role of TPH-1 in the development of CHD, it would be interesting to generate transgenic mice models with a monitored secretion of 5-HT, for example, through the use of TPH-1-KO BON1-bearing nude mice, in order to inhibit 5-HT secretion, or otherwise NET-bearing mice with overexpression of TPH-1 to increase 5-HT secretion and generate a CHD model.

Though other drugs may have anti-diarrheal mechanisms that are unrelated to 5-HT, such as loperamide (an opioid receptor agonist) [80], a significant clinical benefit on gastrointestinal symptoms was shown after the administration of 5-HT3 receptor antagonists, like alosetron and ondansetron, ensuring a decrease in the frequency of diarrhea [81].

Many studies have demonstrated the pivotal role of 5-HT even in development and progression of fibrosis, because it appeared able to stimulate fibroblast growth and fibrogenesis, resulting in CS-associated complications such as CHD, mesenteric and retroperitoneal fibrosis. CHD is a multifactorial phenomenon possibly occurring through 5-HT_2B_R, a G-protein coupled receptor that mediates fibroblasts and smooth muscle cells mitogenic signals, and that induces an increased expression of TGF-β1. In transgenic mice, the overexpression of 5-HT_2B_R, specifically in the heart, resulted in cardiac hypertrophy and extracellular matrix deposition. Specific antagonists of 5-HT_2B_R reduced fibrosis and protected against right ventricle failure in a mice model with pulmonary hypertension [4]. In 2007 Hauso et al. studied the effect of terguride, a 5-HT_2B/2C_R antagonist, on rat models subjected for four months to daily subcutaneous 5-HT injections. Mice treated both with 5-HT and terguride did not show any macroscopic skin changes, heart/liver/stomach weight gain or right-sided echocardiographic changes compared to controls mice treated only with 5-HT, indicating that terguride may prevent 5-HT-induced heart disease. Moreover, terguride had no effect on 5-HT-induced flushing, suggesting that these receptors are not involved in the pathophysiology of flushing in rats [63]. Recently, Waldum et al. confirmed these data, assuming that a specific antagonist could be used in the prevention of progressive valvular disease, eventually in combination with a 5-HT synthesis inhibitor, such as telotristat [82]. At the same time, Roberts et al. described the preclinical pharmacology of the novel 5-HT_3_R partial agonist CSTI-300 in that decreasing the activity of the receptor evoked by the endogenous 5-HT could have a potential role in treating some of the symptoms of CS with a predicted reduced side-effect profile in comparison to telotristat and 5-HT_3_R antagonists. In fact, the 5-HT_3_R partial agonist prevented the complete inhibition of the receptor, reducing the probability of constipation and ischemic colitis. CSTI-300 was shown to be a high-affinity 5-HT_3_R partial agonist with a good dose-dependent efficacy in a rodent model, representing a potential promising treatment mostly for patients with diarrhea-predominant CS, with less constipation than alosetron [83].

Regarding fibrotic complications of the CS, the main factors related to fibrosis development are TGF-β, platelet-derived growth factor (PDGF), basic fibroblast growth factor (FGF2) and connective tissue growth factor (CTGF or CCN2). Therefore, identifying drugs able to affect these pathways could be useful for controlling and reducing this fearful complication [84]. For example, tamoxifen, which inhibits the TGF-β secretion by fibroblasts, and imatinib, which targets the PDGF, should be evaluated more carefully as specific new anti-fibrotic agents. Interestingly, it was noted that imatinib decreased organ fibrosis in scleroderma and pulmonary fibrosis, while Biasco et al. showed a reduction in retroperitoneal fibrosis after a long-term treatment with octreotide and tamoxifen in a 64-year-old man with a NET complicated by fibrosis causing right ureteral obstruction [85].

Angiogenesis inhibitors, tyrosine kinase inhibitors and inhibitors of the fibroblast growth factor pathways are currently being studied in patients with NETs without evaluating their anti-fibrotic effects. Therefore, in the future it would be essential to focus these studies also on the effects of these compounds on tumor microenvironment and growth factors secretion [11].

The development of novel treatments for CS has focused also on oncogenic cellular signaling pathways. Upregulation of the PI3K pathway has been identified as a critical component in the growth and progression of numerous cancer types, including NETs [86,87]. PI3K/Akt signaling is over-activated through either constitutive activation of PI3K, which indirectly promotes Akt phosphorylation and activation, or downregulation and mutations of the endogenous inhibitor of PI3K/Akt signaling PTEN [88]. The PI3K pathway is involved in vesicle trafficking and the secretion of bioactive products, and it is a negative regulator of peptide secretion. Valentino et al. evaluated the role of PI3K and RAS/MEK pathways in NET proliferation, apoptosis and secretion [89]. The study was performed in vitro on several human NET cell lines (BON-1, NCI-H727 and QGP-1) and subsequently in vivo on mouse models. They showed that the PI3K/mTOR inhibitor BEZ235 in combination with the MEK inhibitor PD0325901 decreased neurotensin peptide release, suggesting a potential benefit of PI3K/mTOR/MEK inhibitors in the treatment of patients with NETs and CS. They extended these findings by confirming that combination of BEZ235 + PD0325901 significantly inhibited the growth of BON-1 xenografts in nude mice. Interestingly, the PI3K inhibitor BKM120 also showed in vitro an anti-proliferative activity, but was associated with an increase in neurotensin peptide secretion [89]. Moreover, Silva et al. identified the relationship between PI3K/Akt/PTEN signaling and the secretion and synthesis of 5-HT in BON-1 cells. PTEN is involved in the expression of TPH-1 and secretion of 5-HT. Inhibition of PTEN in vitro, with concomitant increased Akt signaling, resulted in decreased secretion of 5-HT due to reduced expression of TPH-1 [90].

Regarding oncogenic cellular signaling pathways and CHD, mTOR inhibitors were shown to decrease the expression of 5-HT receptors on valve cells, having to be preferred to other cardiotoxic treatments, such as sunitinib, in patients with NETs at risk of developing CHD. The allosteric mTOR inhibitor everolimus was approved by FDA for pancreatic NET treatment, thanks to its ability to target mTOR complex 1 (mTORC1) and its anti-proliferative and anti-angiogenic properties. Unfortunately, the response to everolimus is often not durable in patients with NETs perhaps because of the unsustained inhibition of mTORC1 signaling and/or activation of mTORC2. Therefore, it is mandatory to study alternative and second-line therapies [91]. Orr-Asman et al. tested the preclinical efficacy of mTOR inhibitors in BON-bearing mice, focusing on CC-223, a second generation mTORC1/2 inhibitor. Approximately 57% of BON-luci bearing mice that progressed on rapalog therapy showed a significant decrease in tumor volume upon a switch to CC-223. Mice treated with CC-223 showed decreased cardiac dilation and thickening of heart valves more than those treated with placebo or rapamycin alone [92].

In Figure 1 are reported the main pathways and pharmacological targets with potential future clinical applications.

Clinical evidence also demonstrated that metformin decreased 5-HT secretion in BON-1, but not in QGP-1 cells, emphasizing the intrinsic heterogeneity of NETs. These results could be clinically relevant for patients with CS, because elevated 5-HT levels are directly associated with symptoms [93].

Recently, a retrospective study of 98 NET patients investigated the beneficial effect of an amino acid-based oral rehydration solution which was found to have anti-diarrheal properties in preclinical models. Enterade^®^ AO consists of a unique blend of five amino acids (valine, aspartic acid, serine, threonine, tyrosine), with the ability to restore bowel absorption and integrity, in combination with electrolytes and flavors [94]. Long-term prophylactic use of this solution modulated intestinal transmembrane proteins to promote intestinal villi regrowth, increased sodium and water absorption, decreased chloride and bicarbonate secretion and reduced intestinal paracellular permeability [95]. The results of this study are encouraging and support prospective validation of a non-toxic and inexpensive therapy for effectively treating resistant diarrhea in NET patients [94]. There are two ongoing prospective phase II studies which are currently evaluating the ability of Enterade^®^ AO.

Another pathway to be studied with the intention of identifying new therapeutic targets for CS is represented by the nucleotide-binding oligomerization domain (NOD)-like receptors (NLRs), which are cytoplasmic sensors able to relieve danger signals released by invading pathogens or damaged tissue/cells. Mutations in the NLRP subfamily could affect pro-inflammatory mediators causing non-specific systemic symptoms. In this context, Jacob et al. described an interesting case of a 63-year-old female with abdominal pain, non-bloody diarrhea, and a history of chest pressure, tachycardia, shortness of breath, flushing and hypoglycemia. Symptoms were partially improved with subcutaneous octreotide, and she was diagnosed with undifferentiated connective tissue disorder with positive autoantibodies. Considering that the patient also had an elevated urine 5-HIAA and symptoms concerning for CS, she underwent extensive gastrointestinal workup which showed elevated DOTATATE activity localized to a small focus in the left hepatic lobe, along the lesser curvature of the stomach, tail of the pancreas and small bowel. Meanwhile, genome-level sequencing was requested due to concern for one of the multiple endocrine neoplasms or auto-inflammatory disorders, identifying a heterozygous variant in the *NLRP12* gene. Moreover, the analysis of serum cytokines revealed elevated levels of pro-inflammatory ones. The authors sought to identify a potential genetic etiology of an inflammatory syndrome in a patient with an atypical multisystem illness with CS as well as atopic and autoimmune features. They identified a novel variant in the NLRP12 expanding the spectrum of clinical manifestations attributed to the *NLRP* subfamily gene variants and suggesting a role of NLRP12 in the regulation of multiple cytokines. This variant was found in a patient with symptoms compatible with a CS, so it could be interesting to investigate the potential role of this pathway in the development of CS [96].

Natural zeolites are microporous crystalline aluminosilicates with channels and cavities of molecular dimensions of interest for biomedical applications. A relevant anti-phlogistic effect of natural zeolites has been observed in a murine model of inflammatory disease using 12-O-tetradecanoylphorbol-13-acetate (TPA), as an inflammatory agent. The anti-inflammatory activity seems to be related to the adsorption of histamine on zeolite, as well as siloxane surfaces of smectites, which were effective adsorption sites for amine compounds, such as histamine. The outcome of this work showed in vivo anti-phlogistic properties of zeolite as determined by a murine inflammation model contributing to a better understanding of application in inflammatory bowel diseases and in the topical treatment of inflamed skin with zeolite paste. The inhibitory effects of these compounds on histamine levels may open a new therapeutic scenario in CS. Indeed, the role of amines, like the histamine itself, is relevant in the development of CS. Therefore, starting from this assumption, it would be interesting to investigate in the future the impact of this compound in symptomatic patients with NETs and high levels of histamine [97].

## 4. Future Development of Preclinical Models in Carcinoid Syndrome

The development of efficacious and personalized pharmacological treatment options is strictly related to the choice of preclinical models that faithfully recapitulate the human disease. Both in vitro and in vivo models provide different experimental possibilities to dissect different aspects of CS related to the hormone secretion, tumor microenvironment and clinical manifestations.

The development of in vitro or in vivo platforms of primary NET tissue is one of the most promising and attractive areas of NET research. In the attempt to mimic in vitro some aspects of the complex tumor microenvironment, 3D patient-derived models have been developed, aimed at retaining the 3D architecture of the tissue, providing more faithful representation of stromal and extracellular matrix contributions. Recently, different 3D cultures derived from primary GEP-NET cells have been generated [18,98,99,100]. It is known that NET-associated fibrosis is caused by 5-HT, growth factors and other peptides secreted by NET cells. The crosstalk between these factors and different cellular components of the tumor microenvironment is important in the pathophysiology of the fibrotic process [101]. In this context, primary 3D-culture and co-culture models may represent a useful tool in the elucidation of the complex and dynamic interactions between different cellular populations involved in the onset and progression of CS. In this frame, to investigate the role of 5-HT and growth factors in regulating fibrosis and proliferation in the tumor microenvironment, a co-culture model has been developed with KRJ-I (small intestinal NET cell line) and HEK293 (human embryonic kidney cell line, consisting of endothelial, epithelial and fibroblast cells) cells [102]. In this co-culture model the release of KRJ-I-derived 5-HT played an important role in the regulation of peritumoral fibrosis and angiogenesis, activating the proliferation and fibrogenic activity in HEK293 cell line, underling the important role of 5-HT as mediator in the crosstalk between tumor and stromal cells [102]. It could be interesting to develop new co-culture platforms to investigate the reciprocal influence of two different cell populations, such as carcinoid cell lines and valve interstitial cells. In particular, the analysis of the crosstalk between these cell populations, the growth factor production and 5-HT release could shed new light on the development of CHD [103,104].

Moreover, omic technologies could be combined with 3D culture and 2D co-culture platforms to dissect molecular mechanisms involved in hormone release or growth factor synthesis, with the aim to discover novel biomarkers and therapeutic targets for the CS.

Recently, an ex vivo 3D flow-perfusion bioreactor system has been proposed to culture patient-derived NET cells [105]. This platform could be useful to propagate carcinoid tissues derived from CS patients, considering the difficulties in maintaining NET primary culture for a long time. An interesting application of this tool could be the evaluation of the effects of selected pharmacological treatments aimed at controlling serotonin secretion and monitoring the response over time.

In vivo cancer models often enable the capture of the disease aetiology, progression and resolution processes with the advantage of simulating the complex tumor microenvironment in the context of a whole organism.

Carter A.M. et al. developed a bi-transgenic mouse line in which the aberrant activation of the cyclin-dependent kinase 5 (Cdk5) pathway can selectively induce the growth of pancreatic β-cells with consequent generation of heterogeneous population of NETs [106]. The utility of this model is enhanced by the ability to form tumor-derived allografts in a large cohort of animals. It may represent the starting point for the development of innovative transgenic mouse models of midgut and bronchial NETs with CS, for example, through the transgenic overexpression of *TPH-1* promoter mediated by the activation of Cdk5 pathway. Indeed, this pathway seems to be involved in the tumorigenesis of several NETs [106,107].

Despite rodents representing the gold standard for in vivo studies in cancer research, zebrafish (*Danio rerio*) has emerged as a powerful alternative vertebrate model for the preclinical study of different human diseases. The morphological and functional conservation of neuroendocrine system from fish to mammals, together with the possibility of modelling in zebrafish different diseases with different experimental approaches, make zebrafish particularly versatile also in NET research [108,109]. As in human, the zebrafish neuroendocrine system is composed by anatomically recognizable structures and dispersed cells in different tissues, such as enterochromaffin cells in the intestine that control intestinal movements by 5-HT secretion [108,110]. In addition, the serotonergic system has been well-characterized in the zebrafish adult and larvae.

In the last decades, zebrafish has emerged as a worthwhile in vivo model of human cancer, proving to be similar to the human counterparts both molecularly and pathologically, with the key advantage of the ability to study tumor development and progression at a sophisticated, single-cell level in the context of a simple whole-organism, where tumor cells interact with the microenvironment. In particular, the engraftment of mammalian cancer cells can be performed into different sites of two-day-old larvae at a stage prior to the development of the adaptive immune system, avoiding the implant rejection, or in immunodeficient adult fish [111,112,113]. Zebrafish embryo xenograft platform represents a powerful vertebrate model to visualize, follow and dissect different aspects and mechanisms that drive cancer formation, tumor-induced angiogenesis and tumor cell spread in vivo, in real-time and in a short time-window (within five days post-fertilization). Moreover, by taking advantage of the permeability of zebrafish embryos to small molecules, that can be dissolved in their culture media, it is easy to study the effects of several drugs through this model. Tumor xenograft procedure in zebrafish embryos requires a small number of tumor cells (100 cells/embryo), allowing the possibility to perform patient-derived xenografts (PDXs) with post-surgical samples. This aspect is extremely relevant in NETs, where the size and the availability of tumor cells are often restricted. Zebrafish tumor xenograft platform has already been successfully developed to investigate different hallmarks of NETs, such as angiogenesis, cancer cell spread and metastasis formation, as well as to analyze the effects of several drugs [109,114,115,116,117,118,119,120,121,122]. In this context, BON-1 cells implanted in 48 hpf *Tg(fli1a:EGFP)^y1^* zebrafish embryos, that allow the in vivo visualization of the entire vascular tree [123], showed both pro-angiogenic and invasive behaviors as early as 24 hpi (hours post-injection) and within 48 hpi (Figure 2).

It could be interesting to take advantage of genome editing tools to genetically manipulate carcinoid cells in genes involved in CS development, before their implantation in zebrafish embryos. For example, CRISPR/Cas9 technique could be used in gene-editing of carcinoid cells, as recently described for BON-1 cells [124].

Zebrafish xenograft platform may represent a suitable model to study CS manifestations in different organs, also at molecular level. In future studies, it would be interesting to explore the ability of NET cells, implanted in zebrafish embryos, to develop signs and morpho-functional alterations recapitulating the human CS. Zebrafish represents a good model to study heart development and functionality. Several markers of cardiac tissues and valve have been identified, as well as different heart transgenic lines that have been developed [125,126,127,128]. The use of specific cardiac transgenic zebrafish lines for the implantation of carcinoid cells may enable the in vivo heart visualization with non-invasive techniques, allowing the comparison of cardiac morphology and functionality between controls and implanted embryos. Interestingly, the analysis of heart functionality and phenotype in zebrafish adults can be performed by high frequency ultrasound echocardiography [129], with results useful to identify and monitor in implanted zebrafish over the time the typical signs that affect the heart of CS patients. The formation of fibrotic tissue can be detected and evaluated by histological analysis at the level of involved organs, such as heart and intestine in zebrafish after NET cells implantation. Indeed, several specific histological staining methods for extracellular matrix components and immunofluorescence assays with specific antibodies are available in zebrafish [130].

Finally, Zebrafish/NET cells xenograft platform results are particularly suitable to perform preclinical drug screening [115]. In particular, PDXs with carcinoid tumor cells in zebrafish may represent an innovative platform to perform drug screening and predict drug sensitivity in patients, opening a promising scenario for precision medicine in patients with NETs and CS.

## 5. Conclusions

Preclinical models are indispensable tools for investigating the pathogenesis, pathophysiology, mechanisms for tumor invasion and metastasis and new therapeutic approaches for cancer. Only a few models are available for the study of CS in NETs. A summary of the pros and cons of these models is reported in Table 3.

To date, there are only a few efficacious treatments for CS, especially regarding CHD and mesenteric fibrosis. Therefore, there is a critical need for developing new preclinical models able to detect innovative specific molecular targets and to provide the basis for promising clinical studies.

Although murine models remain the gold standard for preclinical study in NETs, primary 3D-culture, co-culture and zebrafish/tumor xenograft models represent attractive, fast and technically simple tools to study tumor–host microenvironment and to test the antitumor effects of new compounds with potential clinical applications in patients with CS.

Finally, it would be interesting to focus future pharmaceutical research on new compounds targeting oncogenic cell signaling pathways, growth factors and the downstream pathways through which 5-HT acts.

## Figures and Tables

**Figure 1 ijms-24-03610-f001:**
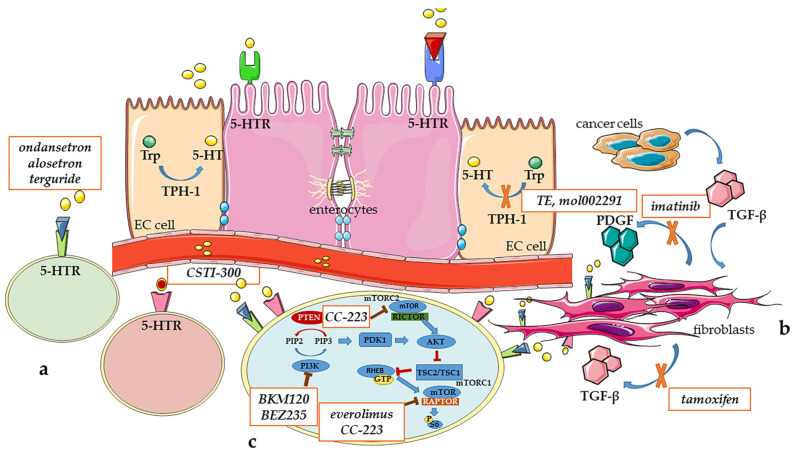
The main pathways involved in carcinoid syndrome and related drugs. (**a**) The main pathway involved in carcinoid syndrome (CS) is related to serotonin (5-HT). Drugs able to block this pathway act through inhibition of tryptophan hydroxylase 1 (TPH1), the isoform expressed in enterochromaffin cells of the gastrointestinal tract (EC cell), or modulation of 5-HT receptors (5-HTRs). TPH1 inhibitors currently studied are telotristat ethyl (TE) and mol002291. Other drugs acting through interaction with 5-HT receptors (5-HTRs) are alosetron and ondansetron (both 5-HT3 receptor antagonists), terguride (5-HT_2B/2C_ receptor antagonist) and CSTI-300 (5-HT3 receptor partial agonist). (**b**) Regarding fibrotic complications of the CS, transforming growth factor beta (TGF-β) can be produced by cancer cells and acts on the fibroblasts by stimulating the production of TGF-β itself. The components of tumor microenvironment induce the expression of platelet-derived growth factor receptor β (PDGFRβ) on fibroblasts and upregulate their activity in a paracrine manner. Tamoxifen and imatinib are able to affect TGF-β and PDGF secretion with potential application in the therapy of CS. (**c**) Considering oncogenic cellular signaling pathways, the pan-PI3K inhibitor, BKM120, and the dual PI3K/mTOR inhibitor, BEZ235, decrease proliferation in multiple NET cell lines. Inhibition of PTEN with concomitant increased Akt signaling decreases secretion of 5-HT due to reduced expression of TPH1. Everolimus targets mTOR complex 1 (mTORC1), but its effect is not durable in patients with NETs because of the unsustained inhibition of mTORC1 signaling and/or activation of mTORC2, that is targeted by CC-223, a second generation mTORC1/2 inhibitor.

**Figure 2 ijms-24-03610-f002:**
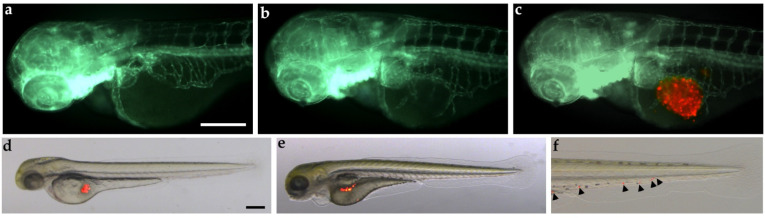
Tumorigenic potential of BON-1 cells implanted in zebrafish embryos. Red-stained BON-1 cells were implanted in 48 hpf *Tg(fli1a:EGFP)^y1^* zebrafish embryos. Epifluorescence images at 24 hpi of PBS-injected embryos (control; (**a**)) and BON-1 (red) xenografted embryos (**b**,**c**). The red channel was omitted in panel b to facilitate the observation of tumor-induced angiogenesis (green). BON-1 xenograft induced the formation of endothelial structures (green) sprouting from the subintestinal vein plexus within 24 h. Overlay of representative fluorescent and bright field images of grafted embryos at 0 (**d**) and 48 hpi (**e**,**f**) showed the spread of tumor cells throughout the embryo body (black arrowhead). The tail particular at 48 hpi was imaged (**f**). Embryos are shown anterior to the left. Scale bar, 100 µm.

**Table 1 ijms-24-03610-t001:** Main features of most used and well-characterized NET cell lines. For each cell line source, biochemical and molecular expression patterns, drug sensitivity/resistance and related references are listed.

Cell Line	Source	Biochemical and Molecular Properties	Drug Sensitivity	References
CGP	Jejunum	Low proliferation rate; synthesis, store, and release of 5-HT and histamine	Unknown	[19]
KRJ-I	Small intestine	Secretion of 5-HT, noradrenaline and pituitary adenylate cyclase; expression of CgA, NSE, Ki-67, TPH-1, substance P and guanylin	High resistance to octreotide-mediated 5-HT secretion inhibition; sensitivity to octreotide + RAD001 co-treatment and AN-238	[20,21,22,23,24]
BON-1	Pancreas	Synthesis of 5-HT, CgA, neurotensin and pancreastatin; expression of gastrin, somatostatin, 5-HT, and acetylcholine receptors; expression of IGF and IGFR	Sensitivity to imatinib, leflunomide, sodium butyrate, hexamethylene bisacetamide, everolimus, pasireotide, LY294002, MK-2206, octreotide, IFN-β and BYL719	[25,26,27,28,29,30,31,32,33,34,35,36,37,38,39]
QGP-1	Pancreas	Expression of TPH-1, CgA, synaptophysin, VMAT1, mGluR4, ADB1, ACM4, substance P, SERT and guanylin	Sensitivity to everolimus, octreotide and BYL719	[33,36,40,41]
NCI-H727	Bronchus (typical)	Expression of p53, SST-2 and SST-5; secretion of calcium-stimulated PTHLP	Sensitivity to EGFR monoclonal antibodies, LY294002, SSA, BYL719, BIM23120 and BIM23A779	[36,42,43,44]
NCI-H720	Bronchus (atypical)	Expression of SST-2 and SST-5	Sensitivity to NSC 95397, brefeldin A, bortezomib, lanreotide, BIM23206 and BIM23120	[45,46]
COLO 320 DM	Colon	Synthesis of 5-HT, parathyroid hormone, ACTH, norepinephrine and epinephrine	Sensitivity to oxaliplatin	[47,48]
GOT1	Ileum	Expression of all recognized somatostatin receptors, VMAT1 and VMAT2	Binding to radiolabeled SSA	[49]

CgA = chromogranin A; NSE = neuro-specific enolase; TPH-1 = tryptophan hydroxylase 1; 5-HT = serotonin or 5-hydroxytryptamine; VMAT1 = ATP-dependent vesicular monoamine transporter 1; mGluR4 = metabotropic glutamate receptor 4; ADB1 = b1-adrenergic receptor; ACM4 = muscarinic 4 acetylcholine receptor; SERT = serotonin transporter; IGF = insulin-like growth factor; IGFR = insulin-like growth factor receptor; PDGFR = platelet-derived growth factor receptor; EGFR = epidermal growth factor receptor; SST-2 = Somatostatin Receptor Type 2; SST-5 = Somatostatin Receptor Type 5; PTHLP = parathyroid hormone-like protein; ACTH = adrenocorticotropic hormone; VMAT2 = ATP-dependent vesicular monoamine transporter 2; SSA = somatostatin analog; IFN-β = interferon β.

**Table 2 ijms-24-03610-t002:** Animal models on carcinoid syndrome (CS). For each study we reported the animal model, its laboratory-molecular and biological properties, the tested drugs and the results achieved.

Animal Model	Laboratory, Molecular and Biological Properties	Tested Drugs	Results	References
GOT1-bearing nude mice	- Increased plasma 5-HT levels and 5-HIAA urine levels. - Tumors express somatostatin receptors and VMAT1 and VMAT2.		All xenografted tumors could be visualized scintigraphically using the SSA 111In-octreotide and 123I-MIBG.	[49]
		177Lu-DOTATATE and then111In-DOTATATE.	Suboptimal therapeutic amounts of 177Lu-DOTATATE caused an increased uptake of the second injection (111In-DOTATATE).	[53]
BON-1-bearing athymic nude mice	- Liver metastases in 65% with elevated platelet 5-HT levels and fibrosis on the valvular tissue (above all on tricuspid valve).			[54]
BON-1- bearing athymic nude mice	- Increased plasma 5-HT levels and 5-HIAA urine levels. - Liver metastasis with positive staining for 5-HT and CgA. - Fibrosis, diarrhea and fibrotic cardiac valvular disease (tricuspid and mitral thickening).	Octreotide/bevacizumab	Octreotide/bevacizumab reduce liver metastasis and manifestation of CS, including valvular heart disease.	[55]
Sprague–Dawley rats		Subcutaneous injections of 5-HT daily for 3 months	- Increased plasma 5-HT levels. - Flushing, loose stools and anorexia. - Cardiac disease with pathological echocardiographs and histopathological changes (shortened and thickened aortic cusps with an increased cellularity of myofibroblasts in a collagenous matrix).	[56]
Sprague–Dawley rats		Subcutaneous injections of 5-HT daily for 7 days	- Higher amount of glycosaminoglycans and a lower amount of collagen. - Thickening and compositional alteration of aortic and mitral valves.	[57]
New Zealand white rabbits		Long term oral administration of 5-HT	- Increased plasma 5-HT levels and 5-HIAA urine levels. - Valvular heart disease with thickened aortic, mitral and tricuspid leaflets and several areas of chondroid metaplasia.	[58]
Transgenic mice overexpressing the Gq-coupled 5-HT_2B_R specifically in the heart	- Proliferation of the mitochondria. - Hypertrophic cardiomyopathy.			[59]
5-HTT-deficient mice	- Increased and persistent interactions between 5-HT and 5-HTR and valvular mitogenic activity with extracellular matrix production. - Structural and functional cardiac abnormalities and valvulopathy.			[60]
Sprague–Dawley rats transplanted intraocularly with midgut carcinoid tumors secreting 5-HT		Application of adrenoceptor agonists locally to the eye	- The activation of beta-adrenoceptors (Isoprenaline) causes release of 5-HT from carcinoid tumor cells. - The stimulation with alpha-adrenoceptors (Norepinephrine) did not elicit any 5-HT release.	[61]
Wistar rats		Oral administration of PCPA and PEPA	Reduced plasma 5-HT levels and 5-HIAA urine levels.	[62]
Sprague–Dawley rats		Subcutaneous injections of 5-HT daily for 4 months and then also with Terguride	- Vasodilatation and decreased heart rate. Block of serotonin-induced changes in the skin (achantosis). - Not heart/liver/stomach weight gain or right-sided echocardiographic changes. - Flushing.	[63]
Transgenic RT2 mice with B6AF1 genetic background	- Loss of imprinting of IGF2 with its overexpression. - Development of ileal NETs. - Secretion of 5-HT in 22% of ileal NETs.		IGF2 is considered as the first genetic driver of ileal neuroendocrine tumorigenesis.	[64]
C57BL/6 and C57 albino mice		Oral administration of LP-920540 and LX1032	- Reduced 5-HT levels in the intestinal mucosa and in plasma. Neither brain 5-HT nor 5-HIAA urine levels were affected significantly. - Improvement of colonic motility.	[65]
C57BL/6 mice models of intestinal inflammation		Oral administration of LX1032/LX1606	- Decreased pro-inflammatory cytokine levels and 5-HT intestinal levels. - Reduced colitis severity and diarrhea episodes.	[66]

CS = Carcinoid syndrome; CgA = Chromogranin A; IBS = irritable bowel syndrome; IGF2 = Insulin Like Growth Factor 2; ^111^In-DOTATATE = ^111^Indium-[DOTA°,Tyr^3^]octreotate; ^177^Lu-DOTATATE = ^177^Lutetium-[DOTA°,Tyr^3^]octreotate; mTORKi = mTOR kinase inhibitors; fenclonine, PCP = p-chlorophenylalanine; PEPA = p-ethynylphenylalanine; SSA = Somatostatin analogue; VMAT = vescicular monoamine transporters; 5-HIAA = 5-hydroxyindoleacetic acid; 5-HT = 5-hydroxytryptamine; 5-HTT = 5-HT transporter; 5-HT_2B_R = 5-HT_2B_ receptor.

**Table 3 ijms-24-03610-t003:** Pros and Cons of preclinical models in carcinoid syndrome.

Model	Pros	Cons
Immortalized NET cell lines	Useful for the study of disease mechanisms and drug efficacy in a controlled environment. Reliable and reproducible results. They provide a means for testing a large number of compounds or interventions in a relatively short time frame.	They may not accurately represent the complexity and heterogeneity of the disease in humans. They may not predict the efficacy or toxicity of a drug in patients with NETs. They may lose characteristics of the original tumor (e.g., secretory function).
Primary cultures of human NETs	A more accurate representation than immortalized cell lines that have been passaged multiple times in culture. Representative of the tumor and the host microenvironment. Cost-effective compared to in vivo models.	Difficult to establish and maintain. High variability. Limited lifespan due to the onset of senescence, which makes them difficult to be used for long-term studies.
Xenotransplantation of NET cells in nude mice	Most of these animals developed liver metastases with elevated platelet 5-HT levels. Most of these animals developed mesenteric fibrosis, diarrhea and fibrotic cardiac valvular disease reminiscent of CS. They provide a more accurate representation of the disease than cell culture models, allowing for the study of the interactions between the tumor and the host and of drug efficacy and toxicity in a whole organism.	They may not accurately represent the complexity and heterogeneity of the disease in humans as the xenografts are grown in an immunodeficient mouse, which does not have a human immune system. Cons related to the injection of immortalized cell lines previously reported. PDX are extremely difficult to be developed, due to the low proliferation rate of primary cultures. Expensive.
Administration of 5-HT in rabbit models or transgenic mouse models with alterations of 5-HT pathway	5-HT injections induced clinical signs observed in patients with CS (flushing, loose stools and anorexia). Long-term 5-HT overload can cause valvular heart disease, similar to that reported in patients with CHD.	They may not accurately represent the complexity and heterogeneity of the disease in humans. They may not predict the efficacy or toxicity of a drug in patients with NETs.
Mouse models of intestinal inflammation	Adopted to test the effects of TPH inhibitors on 5-HT intestinal levels, colitis severity and diarrhea episodes.	They may not accurately represent the complexity and heterogeneity of the disease in humans because they are not specific models of CS/CHD.
Primary 3D-culture and co-culture models	Maintenance of 3D architecture of the tissue, providing a faithful representation of stromal and extracellular matrix contributions. Possibility to study the crosstalk between selected cell populations involved in CS. Possibility to study the secretion of different factors from primary cells derived from CS patients. Possibility to perform drug screening. Omic technologies could be combined with these platforms to dissect specific molecular mechanisms, with the aim to discover novel biomarkers and therapeutic targets for the CS.	Difficulties in cell culture due to the low proliferation rate of primary NET cells. Limited representation of the disease due to the loss of interactions between tumor cells and the whole organism. Models to be validated for CS.
Zebrafish xenograft models	Tumor implant with low number of cells. No tumor cell rejection in embryos and larvae due to the lack of a fully developed immune system. High number of embryos are available for each experiment. High engraftment rate. In vivo and real-time visualization of tumor implants. Embryo tissue permeability to small molecules, useful for drug-screening.	Differences between the optimal temperature for zebrafish development and mammalian cell metabolism. Difficulties in recapitulating all clinical CS manifestations. Difficulties for long-term analyses. Difficulties in administration of molecules with low water solubility. Models to be validated for CS.

CS: carcinoid syndrome. CHD: carcinoid heart disease. 5-HT = 5-hydroxytryptamine. PDX: patient-derived xenografts. TPH: tryptophan hydroxylase.

## Data Availability

Not applicable.

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
