# Peer review of "Carcinoid Syndrome: Preclinical Models and Future Therapeutic Strategies"

_ijms, 2023, doi:10.3390/ijms24043610_

Round 1

Reviewer 1 Report

The review aims to inform about models that are valuable in researching NETs and carcinoid syndrome. Available methods and their individual characteristics and uses are described thoroughly and in detail. The review would benefit from further discussion on the pros and cons of using these models, their strengths but particularly their limitations

Author Response

Comments and Suggestions for Authors

The review aims to inform about models that are valuable in researching NETs and carcinoid syndrome. Available methods and their individual characteristics and uses are described thoroughly and in detail. The review would benefit from further discussion on the pros and cons of using these models, their strengths but particularly their limitations

Answer: as suggested by the reviewer we have underlined the pros and cons of these preclinical models, summarized in table 3.

Reviewer 2 Report

This is a very good topic for a review. However, in its present form, it reads more like grant application for funding to model carcinoid syndrome in zebrafish. The zebrafish section is overly long for a model of CS that does not even exist.

Statements are often made without providing references for them. Other references that are key to this topic are simply missing.  

For instance, after reading this review, I was still left with many very basic questions about Carcinoid Syndrome including:

1.     Just how common is Carcinoid Syndrome? I googled this, and found a range of values, mostly as a percentage of the total number of carcinoid cases. I suspect that there’s no well-defined number. Regardless, as a review on CS, this review should include such information.

2.     Among patients with CS, just how common is CHD? Isn’t this a very rare event even among patients with CS? Any idea about how many patients will die or become debilitated by CHD derived from CS?

3.     Is the frequency of Carcinoid Syndrome increasing, as was suggested by Halperin et al in Lancet Oncol 2017, 525-34?  And why isn’t this paper even in the reference list? Should we think that the Halperin study overstates the increase in CS due to small sample size, due to increased education of providers about CS, or even due to a change in the way that CS has been defined over the years definitions for CS over the years? Please discuss or at least reference this.  

4.     Some mention was made about adverse events that accompany treatment with TPH1 inhibitor, but surprisingly these were not referenced. In particular a recent paper by Horsch et al in Neuroendocrinology 112: 298-309 should be discussed.

5.     As far as I am aware, there is no data about whether teloristat treatment reduces the risk or moderates the effect of CHD in patients. But I do see that there is a clinical trial in progress to address this (see https://clinicaltrials.gov/ct2/show/NCT04810091); please mention this in your review.

Other points:

1.     I would expand the discussion of CS models in mice, because they seem quite useful but underexploited for study of CS. In particular I would point out several of the elementary questions that could easily be asked using these models. For instance:

a.     As mentioned above, there is no data on the effect of teloristat on carcinoid heart disease in patients. Couldn’t this be tested using the BON1 xenograft model? Couldn’t the new generation TPH1 inhibitor listed by these authors (mol002291) also be tested for impact on CHD using this approach?

b.     I’m curious that past researchers have generally only used BON1/nude models?  Should one cell line be trusted? Why haven’t previous authors been forced to repeat the same analysis at least using H727/nude models, and now also with GOT1/nude models since GOT1 has become more available?  

c.     From the 5HT injection models, it appears that 5HT is sufficient to generate CHD. But why not repeat this in xenograft mouse models, for instance by knocking out TPH1 in BON1 before xenografting the cell line? Or by raising expression of TPH1 in QGP1, which normally doesn’t make very much 5HT at all.  Perhaps other hormones secreted by these lines would also stimulate CHD?

2.     Eliminate the Drosophila paragraph completely. There is no model for NETs in Drosophila, so there is nothing to review on this topic.

3.     Keep the zebrafish discussion, since I do see the value of using this model for genetic and pharmacological analysis of CS and since there are at least several papers in which different types of NETs have been modeled by zebrafish xenografts. But I’d also shorten this section, since none of the zebrafish models have yet been used to model CS.

4.     It’s not always clear when this review is discussing the modeling of treatment of carcinoid syndrome, as opposed to the modeling of potential anti-tumor treatments. For instance, the in vitro models (bioreactor; 3D) seem less about CS and more about anti-tumor studies.

Minor points:

1.     Could you be more specific about what you mean by “liver-directed therapies” (lines 66 and 67)? Probably you mean embolization, and possibly surgical resection?

2.     I would switch the order of sections 3 and 4. In section 4, you make a pretty good argument for why additional genetic and pharmacological studies could be helpful to better understand this disease; in section 3, you explain why zebrafish and perhaps new 3D models could be used to address some of these questions. In section 3, I’d also include the possibility of manipulating BON1/H727/GOT1 genetically (CRISPR of specific genes or perhaps CRISPR libraries) prior to implanting them.

3.     There is discussion of the cell line CNDT2, including the fact that it derived from a 58 year old patient with an ileal NET. I found a paper in which the developer of this cell line admitted that this patient was likely not the source of this cell line (Ellis et al, Clin Cancer Res 16, 5365-6), and that the cell line may not be a carcinoid-derived cell line at all. This should be included in the review so that other researchers can be forewarned about the dubious origin of CNDT2.

4.     The QGP1 cell line appears to make somatostatin, not serotonin. So wouldn’t this make it a poor model for carcinoid? Maybe this should be stated in the review.

5.     I would drop the long discussion of the krj1/293 coculture model. 293 is neither heart nor even fibroblast; more importantly, krj1 is a cell line that is generally unavailable so these data cannot be tested independently. It’s OK to mention the model as an example of coculture, but I’d suggest how it could be improved greatly.

6.     I would add the work of Ear et al, J Vis Exp 2019, 152 to the in vitro models.  

7.     It’s been hard to generate serotonin-producing genetically engineered mouse models, but Contractor et al finally generated an ileal NET GEMM that produces serotonin (Endocr Rel Cancer 27 175-86). Please reference. Also there’s still no lung carcinoid GEMM but Bibb’s lab seems to be able to generate a variety of different types of NETs by NET-specific expression of CDK5 (eg see Oncogenesis 10; 83 for the latest one), so perhaps a transgenic fusion between TPH1 promoter and CDK5 gene could also be used to generate a lung and/or ileal carcinoid model in mice.  

Author Response

Comments and Suggestions for Authors

This is a very good topic for a review. However, in its present form, it reads more like grant application for funding to model carcinoid syndrome in zebrafish. The zebrafish section is overly long for a model of CS that does not even exist.

Statements are often made without providing references for them. Other references that are key to this topic are simply missing.  

Answer: Following the reviewer’s suggestions we revised our manuscript accordingly (shortening the section on zebrafish and providing references where missing).

For instance, after reading this review, I was still left with many very basic questions about Carcinoid Syndrome including:

  1. Just how common is Carcinoid Syndrome? I googled this, and found a range of values, mostly as a percentage of the total number of carcinoid cases. I suspect that there’s no well-defined number. Regardless, as a review on CS, this review should include such information.
  2. Among patients with CS, just how common is CHD? Isn’t this a very rare event even among patients with CS? Any idea about how many patients will die or become debilitated by CHD derived from CS?
  3. Is the frequency of Carcinoid Syndrome increasing, as was suggested by Halperin et al in Lancet Oncol 2017, 525-34?  And why isn’t this paper even in the reference list? Should we think that the Halperin study overstates the increase in CS due to small sample size, due to increased education of providers about CS, or even due to a change in the way that CS has been defined over the years definitions for CS over the years? Please discuss or at least reference this.  

Answer (1-3): We would like to thank the reviewer for these suggestions. We agree with reviewer that it is very difficult to extrapolate epidemiological data on CS and CHD in NETs due to small series and potential bias of several studies. Additional data on CS and CHD have been reported in the introduction.

  1. Some mention was made about adverse events that accompany treatment with TPH1 inhibitor, but surprisingly these were not referenced. In particular a recent paper by Horsch et al in Neuroendocrinology 112: 298-309 should be discussed.

Answer: as suggested by the reviewer, we discussed the study by Horsch et al. in the “Future Therapies for Carcinoid Syndrome” section.

  1. As far as I am aware, there is no data about whether teloristat treatment reduces the risk or moderates the effect of CHD in patients. But I do see that there is a clinical trial in progress to address this (see https://clinicaltrials.gov/ct2/show/NCT04810091); please mention this in your review.

 Answer: as suggested by the reviewer, we mentioned this clinical trial in progress in the “Future Therapies for Carcinoid Syndrome” section.

Other points:

  1. I would expand the discussion of CS models in mice, because they seem quite useful but underexploited for study of CS. In particular I would point out several of the elementary questions that could easily be asked using these models. For instance:

a) As mentioned above, there is no data on the effect of teloristat on carcinoid heart disease in patients. Couldn’t this be tested using the BON1 xenograft model? Couldn’t the new generation TPH1 inhibitor listed by these authors (mol002291) also be tested for impact on CHD using this approach?

Answer: as suggested by the reviewer, we discussed the potential use of BON1 xenograft model for testing the impact of new drugs on CHD in the “Future Therapies for Carcinoid Syndrome” section.

b) I’m curious that past researchers have generally only used BON1/nude models?  Should one cell line be trusted? Why haven’t previous authors been forced to repeat the same analysis at least using H727/nude models, and now also with GOT1/nude models since GOT1 has become more available?

Answer: We thank the reviewer for this observation. There are only few papers using GOT1 cells implanted in nude models, as reported in the text (Grozinsky-Glasberg, 2012; Kolby, 2001; Bernhardt, 2007). We have also included few papers on H727/nude models (Fu, 2022; Johnbeck, 2014; Petersen, 2012). No information are available on the development of fibrosis and diarrhea and increased production of 5-HT in animals after the implantation of NCI-H727 cells. We believe that the preferred use of BON1/nude models is related to the optimal representation of this disease. Indeed, BON1/nude models developed liver metastasis with increased 5-HT plasma levels, mesenteric fibrosis, diarrhea, and fibrotic cardiac valvular disease. Although BON1 cells are widely used as a model for NET research, it is important to consider that they derived from a specific subtype of tumor and may not accurately represent the complexity and heterogeneity of other subtypes of NETs. In addition, cell lines may not accurately represent the complexity and heterogeneity of the tumor in vivo. We have discussed pros and cons of models in the new table 3.

c) From the 5HT injection models, it appears that 5HT is sufficient to generate CHD. But why not repeat this in xenograft mouse models, for instance by knocking out TPH1 in BON1 before xenografting the cell line? Or by raising expression of TPH1 in QGP1, which normally doesn’t make very much 5HT at all.  Perhaps other hormones secreted by these lines would also stimulate CHD?

Answer: We thank the reviewer for the suggestion. We provided some context in the “Future Therapies for Carcinoid Syndrome” section.

  1. Eliminate the Drosophila paragraph completely. There is no model for NETs in Drosophila, so there is nothing to review on this topic.

Answer: According to the reviewer suggestion, we eliminated the paragraph about the Drosophila model.

  1. Keep the zebrafish discussion, since I do see the value of using this model for genetic and pharmacological analysis of CS and since there are at least several papers in which different types of NETs have been modeled by zebrafish xenografts. But I’d also shorten this section, since none of the zebrafish models have yet been used to model CS.

Answer: According to the reviewer suggestion, we shortened and revised this section.

  1. It’s not always clear when this review is discussing the modeling of treatment of carcinoid syndrome, as opposed to the modeling of potential anti-tumor treatments. For instance, the in vitro models (bioreactor; 3D) seem less about CS and more about anti-tumor studies.

 Answer: We changed some sentences, focusing the attention on CS.

Minor points: 

  1. Could you be more specific about what you mean by “liver-directed therapies” (lines 66 and 67)? Probably you mean embolization, and possibly surgical resection?

Answer: We have specified liver-directed therapies as radiofrequency ablation, cryoablation, transarterial embolization, chemoembolization and radioembolization.

  1. I would switch the order of sections 3 and 4. In section 4, you make a pretty good argument for why additional genetic and pharmacological studies could be helpful to better understand this disease; in section 3, you explain why zebrafish and perhaps new 3D models could be used to address some of these questions. In section 3, I’d also include the possibility of manipulating BON1/H727/GOT1 genetically (CRISPR of specific genes or perhaps CRISPR libraries) prior to implanting them.

Answer: According to the reviewer suggestion we have switched the order of sections 3 and 4. In the discussion about the zebrafish xenograft model, we have added a sentence about the possibility to implant genetically manipulated carcinoid cells.

  1. There is discussion of the cell line CNDT2, including the fact that it derived from a 58 year old patient with an ileal NET. I found a paper in which the developer of this cell line admitted that this patient was likely not the source of this cell line (Ellis et al, Clin Cancer Res 16, 5365-6), and that the cell line may not be a carcinoid-derived cell line at all. This should be included in the review so that other researchers can be forewarned about the dubious origin of CNDT2.

Answer: We would like to thank the reviewer for this comment. The issue raised by Ellis in the up cited letter has been reported. We included a short paragraph about this debate.

  1. The QGP1 cell line appears to make somatostatin, not serotonin. So wouldn’t this make it a poor model for carcinoid? Maybe this should be stated in the review.

Answer: Doihara et al. showed that QGP-1 cells release 5-HT via TRPA1 activation and secrete substance P and guanylin (Doihara et al. QGP-1 cells release 5-HT via TRPA1 activation; a model of human enterochromaffin cells. Mol Cell Biochem 2009;331:239-45). In addition, cabergoline/BIM-065 treatment decreased serotonin release in QGP-1 cells (Endocr Relat Cancer 2019 Jun;26:585-599). Therefore, we believe that their secretory capacity makes these cells a good model of carcinoid syndrome.

  1. I would drop the long discussion of the krj1/293 coculture model. 293 is neither heart nor even fibroblast; more importantly, krj1 is a cell line that is generally unavailable so these data cannot be tested independently. It’s OK to mention the model as an example of coculture, but I’d suggest how it could be improved greatly.

Answer: We shortened the discussion about the mentioned co-culture model, and we rephrased the sentence about the example of a co-culture model with carcinoid cells and valve interstitial cells.

  1. I would add the work of Ear et al, J Vis Exp 2019, 152 to the in vitro models. Answer: We thank you the reviewer for this suggestion. We included the up-mentioned paper in the reference section (PMID: 31657801).  

  1. It’s been hard to generate serotonin-producing genetically engineered mouse models, but Contractor et al finally generated an ileal NET GEMM that produces serotonin (Endocr Rel Cancer 27 175-86). Please reference. Also there’s still no lung carcinoid GEMM but Bibb’s lab seems to be able to generate a variety of different types of NETs by NET-specific expression of CDK5 (eg see Oncogenesis 10; 83 for the latest one), so perhaps a transgenic fusion between TPH1 promoter and CDK5 gene could also be used to generate a lung and/or ileal carcinoid model in mice.  

Answer: As suggested by the reviewer, we discussed both papers in the “In Vivo Models” section.

Round 2

Reviewer 2 Report

It is much better.  Please make a few more changes:

Line 40. Neuroendocrine tumors (NETs) occur in about 7 out of every 100,000 people. While true, this first sentence is also misleading. Yes there are 7 cases of NETs per 100,000; however, since carcinoid syndrome (the subject of the current review) develops mostly from patients with small intestinal NETs, the incidence of SI-NETs should be noted instead of the incidence of all NETs. From the same reference, the incidence of SI-NETs is about 1/100,000. It would be better to insert this number instead of 7/100,000. 

Line 91: "isolated from patients with CS" should be removed. The CS phenotype was not listed for all of the patients who donated the tumors that became these cell lines

Line 168-9: remove  "derived from liver metastasis of a 58-year-old female patient with ileal carcinoid." It's not clear what patient donated this tumor but one thing that we know for certain is that it did NOT come from this 58 year old patient with ileal carcinoid. For evidence, please read the words of the original developers of CNDT2.5 in their letter (reference 54): "unfortunately, we were unable to match our short tandem repeat genotype to the carcinoid tumor that we thought was the source of this cell line". Probably the postdoc mislabeled a patient with a colorectal tumor for a patient with a midgut tumor. Again, I strongly suggest that these authors use this review to point out to researchers that CNDT2.5 really should not be used to model CS. 

Line 313. This is a good place to add a reference to the second genetic driver of Ileal NETs, Mir1-2 (Oncogenesis 9, 37). 

Lines 314-320. I'd move this to section 4, as a future possibility. Also add reference to Cancer Cell 24, 499-511; combined with the more recent paper, this shows that CDK5 can drive both pancreatic NETs and thyroid NETs. Thus for CS research, the question is whether CDK5 can also be used to engineer mice to drive midgut NETs or bronchial NETs, which might then lead to CS.  

Lines 493-6. As written, this is quite confusing. Maybe remove the part about BKM120 and just talk about BEZ235 + PD0325901. Also, clarify which study is being referenced for BEZ235 + PD0325901. Later you can state that BKM120 did slow growth but actually increased peptide synthesis. 

Lines 509-10.  Unfortunately, the response to everolimus is often not durable in patients with NETs because of the unsustained inhibition of mTORC1 signaling and/or activation of mTORC2. Is there a direct reference for this? Otherwise, put "perhaps" in front of the word "because. "

Author Response

Line 40. Neuroendocrine tumors (NETs) occur in about 7 out of every 100,000 people. While true, this first sentence is also misleading. Yes there are 7 cases of NETs per 100,000; however, since carcinoid syndrome (the subject of the current review) develops mostly from patients with small intestinal NETs, the incidence of SI-NETs should be noted instead of the incidence of all NETs. From the same reference, the incidence of SI-NETs is about 1/100,000. It would be better to insert this number instead of 7/100,000. 

Answer: We changed the sentence accordingly.

Line 91: "isolated from patients with CS" should be removed. The CS phenotype was not listed for all of the patients who donated the tumors that became these cell lines

Answer: According to the reviewer suggestion, we removed the sentence.

Line 168-9: remove  "derived from liver metastasis of a 58-year-old female patient with ileal carcinoid." It's not clear what patient donated this tumor but one thing that we know for certain is that it did NOT come from this 58 year old patient with ileal carcinoid. For evidence, please read the words of the original developers of CNDT2.5 in their letter (reference 54): "unfortunately, we were unable to match our short tandem repeat genotype to the carcinoid tumor that we thought was the source of this cell line". Probably the postdoc mislabeled a patient with a colorectal tumor for a patient with a midgut tumor. Again, I strongly suggest that these authors use this review to point out to researchers that CNDT2.5 really should not be used to model CS. 

Answer: as suggested by the reviewer, we removed the sentence in the text and the cell line from table 1

Line 313. This is a good place to add a reference to the second genetic driver of Ileal NETs, Mir1-2 (Oncogenesis 9, 37). 

Answer: as suggested by the reviewer, reference has been cited.

Lines 314-320. I'd move this to section 4, as a future possibility. Also add reference to Cancer Cell 24, 499-511; combined with the more recent paper, this shows that CDK5 can drive both pancreatic NETs and thyroid NETs. Thus for CS research, the question is whether CDK5 can also be used to engineer mice to drive midgut NETs or bronchial NETs, which might then lead to CS.  

 Answer: We would like to thank the reviewer for these suggestions. We moved these sentences in section 4. New reference has been cited.

Lines 493-6. As written, this is quite confusing. Maybe remove the part about BKM120 and just talk about BEZ235 + PD0325901. Also, clarify which study is being referenced for BEZ235 + PD0325901. Later you can state that BKM120 did slow growth but actually increased peptide synthesis. 

 Answer: We changed these sentences accordingly.

Lines 509-10.  Unfortunately, the response to everolimus is often not durable in patients with NETs because of the unsustained inhibition of mTORC1 signaling and/or activation of mTORC2. Is there a direct reference for this? Otherwise, put "perhaps" in front of the word "because. "

Answer: We changed the sentence accordingly.